# Chemical Composition and Antibacterial Activity of *Lippia multiflora* Moldenke Essential Oil from Different Regions of Angola

**DOI:** 10.3390/molecules26010155

**Published:** 2020-12-31

**Authors:** Nsevolo Samba, Radhia Aitfella-Lahlou, Mpazu Nelo, Lucia Silva, Rui Coca., Pedro Rocha, Jesus Miguel López Rodilla

**Affiliations:** 1Chemistry Department, University of Beira Interior, 6201-001 Covilhã, Portugal; ra.aitfella@ubi.pt (R.A.-L.); mpanzu.nelo@ubi.pt (M.N.); mlas@ubi.pt (L.S.); rdcc77@gmail.com (R.C.); np65vo@gmail.com (P.R.); 2Department of Clinical Analysis and Public Health, University Kimpa Vita, Uige 77, Angola; 3Laboratory of Valorisation and Conservation of Biological Resources, Biology Department, Faculty of Sciences, University M’Hamed Bougara, 35000 Boumerdes, Algeria; 4Fiber Materials and Environmental Technologies (FibEnTech), University of Beira Interior, 6201-001 Covilhã, Portugal

**Keywords:** *Lippia multiflora* Moldenke, Angola, GC/MS, antibacterial activity, vapour phase test, agar wells technique, PCA, HCA, heat-map

## Abstract

The purpose of the study was to determine the chemical composition and antibacterial activity of *Lippia multiflora* Moldenke essential oils (EOs) collected in different regions of Angola. Antibacterial activity was evaluated using the agar wells technique and vapour phase test. Analysis of the oils by GC/MS identified thirty-five components representing 67.5 to 100% of the total oils. Monoterpene hydrocarbons were the most prevalent compounds, followed by oxygenated monoterpenes. The content of the compounds varied according to the samples. The main components were Limonene, Piperitenone, Neral, Citral, Elemol, p-cymene, Transtagetone, and Artemisia ketone. Only one of the eleven samples contained Verbenone as the majority compound. In the vapour phase test, a single oil was the most effective against all the pathogens studied. The principal component analysis (PCA) and hierarchical cluster analysis (HCA) of components of the selected EOs and inhibition zone diameter values of agar wells technique allowed us to identify a variability between the plants from the two provinces, but also intraspecific variability between sub-groups within a population. Each group of essential oils constituted a chemotype responsible for their bacterial inhibition capacity. The results presented here suggest that Angolan *Lippia multiflora* Moldenke has antibacterial properties and could be a potential source of antimicrobial agents for the pharmaceutical and food industry.

## 1. Introduction

*Lippia*, dedicated to Augustine Lippi (1678–1701) [1], is a genus of typical flowering plants belonging to the family Verbenaceae [2,3]. The group includes about 200 species of grasses, shrubs and small trees [4], mainly distributed in South and Central American countries and tropical African territories [3,5]. In the Americas, *Lippia* species are found in the arid southwestern United States, in the deciduous rainforests of Central America and the rock and cerrado fields of Brazil, regions with a high index of endemism, extending as far as Uruguay and Central Argentina [6,7,8].

Besides, they are present in subtropical Africa, from East Africa to South Africa [3,9]. Some also coincide with centres of high endemism, in the eastern region, associated with the highest mountains and alpine peaks [8]. The genus *Lippia* is of great economic importance due to the different uses of these essential oils, as it includes several aromatic medicinal species [10,11,12,13,14,15]. Among those species, there is *Lippia multiflora* Moldenke, otherwise known as *Lippia adoensis* Hochst. It is a herbaceous plant that grows in wooded savannahs where it colonises the bush and roadsides [16,17]. The plant is widespread in West and Sub-Saharan Africa, but also South and Central America [5,18,19]. The species is a hardy, woody, the perennial and aromatic shrub that can reach a height of 4.0 m. It has large, oblong, lanceolate bluish-green leaves and whitish flowers on conical heads in a terminal panicle [20].

Known as bush tea, a healing herb, Bunsurun fadama or “godon kada” (Hausa), *L*. *multiflora* Mold. is used in the composition of some traditional improved African medicines, Malarial^®^ in Mali and Tetra^®^ in Congo, and is used to treat various diseases [21,22,23]. It has hypotensive and diuretic properties and relieves fatigue [24]. In folk medicine, it has widely used in the treatment of liver failure, jaundice, stomach aches, lung infections, fever, and buccal candidiasis [5,23,25,26,27]. In Côte d’Ivoire, *L. multiflora* Mold. is commonly used as a substitute for a drink, tea, and as a condiment [5,28]. *L. multiflora* Mold. essential oils show an intraspecific variation in its composition [18,29,30]. The species contains a wide range of chemotypes with different bioactivities and organoleptic profiles. Several studies have demonstrated its sedative, anticonvulsive, anti-infectious, antiparasitic, and antitussive properties [31,32,33,34,35,36]. It has analgesic, antipyretic and anti-inflammatory proprieties [30,37,38]. According to several studies, *L. multiflora* Mold. plant extracts have also shown well-established biological effects. The aqueous extract of *L. multiflora* Mold has vasodilating, hypotensive and cardio-moderating effects [31,39,40,41]. It provides significant protection against ethanol-induced toxicity in rat livers [42]. Furthermore, other types of extracts and their phytochemicals showed antioxidant, antimicrobial, antiplasmodial, and anti-inflammatory activities [43,44].

Essential oils could play a very important role in bacterial infection control programs in the future. They could offer tremendous opportunities for health sectors to develop plant-based compounds against microorganisms and other related problems. There are several reports on essential oils from *Lippia multiflora* Moldenke of multiple origins. Still, to our knowledge, no studies have been conducted on the extraction of essential oils from the Angolan species. The purpose of the present study was to investigate the chemical composition and potential antibacterial activity of *L. multiflora* Mold. essential oils from different regions in Angola. Moreover, the correlation between essential oils components and their antibacterial ability were also evaluated.

## 2. Materials and Methods

### 2.1. Region of Study

Angola is the largest country in Southern Africa (1,246,700 km^2^) [45]. It is bordered to the west by the Atlantic Ocean and the east by Zambia (Figure 1). It shares its borders with Congo in the north and the Democratic Republic of Congo (DRC) in the south with Namibia [46]. The country has more than 32 million population and is divided into eighteen provinces with a wide range of ecosystems and habitats [47]. Indeed, Angola lies between and within two major terrestrial biogeographic regions: the moist forests and savannas of the Congolian region; and the woodlands, savannas, and floodplains of the Zambian territories [48]. Our study was conducted in [49].

The municipality of M’banza Kongo (6°16′00″ S 14°14′00″ E) and Tomboco (6°48′ 33.318″ S 13°17′56.904″ E), located in the province of Zaire, in the north of Angola. The municipality of Cazengo (9°18′ S 14°54″ E) of the province Cuanza Norte located in the north-central region of the country (Figure 1).

### 2.2. Plant collection

The aerial part of *Lippia multiflora* Moldenke (Figure 2) was collected from January 2017 to September 2018 during the three different stages of the plant: L4/L5/L7/L8a/L9a have been sampled before flowering, L1/L2/L8b/L9b during flowering, and L3/L6 after flowering (Table 1). The plants were dried at room temperature in a place protected from direct sunlight and stored in bags. A reference specimen was deposited in the laboratory of Fibber Materials and Environmental Technologies (FibEnTech), University of Beira Interior, Covilhã, Portugal.

### 2.3. Extraction of Essential Oil

*Lippia multiflora* Moldenke essential oil was extracted using hydrodistillation method with Clevenger apparatus following European Pharmacopoeia [50]. After two hours of extractions, EO collected from each plant was then dehydrated with Na_2_SO_4_ and stored at 4 °C until further analysis. The yield of the essential oil was calculated in % (*v*/*w*), based on dry plant weight.

### 2.4. G.C.–M.S. Analysis

The GC/MS analyses were performed with Agilent technology 7890A apparatus equipped with column J&W DB5-ms (30 m × 0.25 mm i.d, film thickness 0.25 m) associated with an MS A 5975C inert XLMSD mass spectrometer. Transfer-line temperature 250 °C, ionization voltage 70 eV. The oven temperature was programmed isothermal at 60 °C for 5 min, then gradually increased to 250 °C at 10 °C/min, held isothermal at 250 °C for 15 min and finally raised to 280 °C at 10 °C/min, injector temperature 250 °C, source temperature 230° C, interface temperature 280 °C, quadrupole temperature 180 °C, carrier gas He (1.0 mL/min), automatic injection volume 1 μL, diluted samples (in dichloromethane) spitless injector. The identification of the compounds was based on a comparison of their retention indexes determined relative to the retention time of aliphatic hydrocarbons (C9–C28) and the mass spectra with those of authentic compounds using NIST databases and Wiley spectral libraries [51,52].

### 2.5. Preparation of the Microbial Strains

Bacterial strains of *Staphylococcus aureus* (ATCC 25923/Lot 902840), *Escherichia coli* (ATCC 25922/Lot 931370) and *Pseudomonas aeruginosa* (ATCC 27853/Lot 931372) were used in this research. The microorganisms were supplied from BR-Ambient and Food Laboratory Lda., of the American Type Culture Collection (ATCC), distributed by Culti-loops^®^ (OXOID Ltd, Basingstoke, UK.) Organisms were maintained on Mueller-Hinton agar (MH2-Ref. 43301; BioMérieux SA, Lyon, France). The microbial suspensions turbidity was made based on 0.5 McFarland (~10^8^ CFU/mL) [50].

### 2.6. Antimicrobial Vapour Phase Test

A rapid vapour phase test determined the antimicrobial activity of the different essential oils according to the method of Lisin et al. (1999) [53]. This technique was conducted using a suspension with a standard McFarland turbidity of 0.5. After filling the Petri dishes with the MH2 culture medium, the bacterial suspension was inoculated uniformly onto agar using a sterile swab and left to stand for 15 min. Then 70 µL of oil was added to fill the lid. Petri dishes containing DMSO 5% or left untreated were used as solvent and growth controls. The plates were inverted and incubated at an average temperature of 37 °C for 24 h. At the end of the incubation period, the effect of the oil on bacterial growth was noted in terms of inhibition zone diameter (IZD) values (mm). This process was carried out three times under sterile conditions.

### 2.7. Well Diffusion Agar

The evaluation of the antimicrobial activity was conducted by the method of agar solid diffusion by perforation of the cylindrical cavities [54]. Briefly, the organisms were further cultivated on nutrient broth at 37 °C for 24 h. The technique was conducted using a suspension with a standard McFarland turbidity of 0.5. L1–L9 essential oils were prepared in a suitable solvent dimethyl sulfoxide (DMSO 5%) and sterilised by a 0.22 μm syringe filter. Wells, 4mm high and 5 mm in diameter, were dug into the MH2 plates (55 mm). They were filled with 70 µL of the raw sample, 1:2 and 1:10 dilution of essential oil. Negative control (DMSO 5%), and two positive controls Penicillin (0.05 mg/mL), Gentamicin A (10 mg/mL) were also tested. The Petri dishes were incubated at 37 °C for 24 h. The antimicrobial effect was done in triplicate and determined in terms of IZD in mm.

### 2.8. Determination of Minimum Inhibitory Concentration (MIC) and Minimum Bactericidal Concentration (MBC)

To determine the MIC and MBC, the macro dilution technique in MHB broth (Muller-Hinton Broth, Ref 724245-Oxoid Ltd, Basingstoke, UK) supplemented with 5% DMSO was used. In one tube, 0.5 mL of inoculum (5 × 10^5^ CFU/mL) prepared in MHB broth (+5% DMSO) and 0.4 mL of MHB broth (+5% DMSO) were placed. Sequential dilutions of EOs were prepared 1:2, 1:4, 1:8, 1:16, 1:32 and 1:64 respectively in DMSO. Two bactericidal antibiotics were used as positive controls, Penicillin G and Gentamicin A at concentrations of 0.05 mg/mL and 10 mg/mL respectively. DMSO was used as a negative control. After 18 h of incubation at 37 °C, 1 mL of resazurin was added. After 30 min, the tubes were visually examined. The change in the colour corresponding to the transformation of resazurin to resofurin reflects bacterial growth.

The MIC values for each EO corresponding to the highest dilution that does not present any visible disorder to the naked eye were recorded. The MBC was determined by quantitative subculturing. A volume of 0.1 mL of each test tube (including controls) was spread on MH2. These plates were incubated at 37 °C. After incubation, colony-forming units (CFU) were counted, and all tests were performed in triplicate. The MBC is defined as the lowest concentration of EO that kills the bacteria tested at 99.9 to 100%.

### 2.9. Statistical Analysis

For GC-MS analysis, the oil components were subjected to hierarchical cluster analysis (HCA) and principal component analysis (PCA). In the case of HCA, the dendrogram (tree) was produced using Ward’s method of hierarchical clustering with squared Euclidean distance between oil samples (with 95% confidence). The cluster analysis (CD) was carried out using the Euclidean distance and the Unweighted Pair Group Method with Arithmetic Mean cluster algorithm. The determination of the effect of *Lippia multiflora* Moldenke on the growth of *Staphylococcus aureus*, *Escherichia coli* and *Pseudomonas aeruginosa* was conducted in three replications. One-way analysis of variance (ANOVA) followed by Tukey’s multiple comparisons test was used to determine significant differences between the means of the inhibition zone diameter. Significance for all tests was determined at a *p*-value of ≤0.05. HCA, CD and PCA also studied the correlation between the essential oil chemotypes and antibacterial activity. All statistical analyses were performed using IBM SPSS Statistics for Windows (Version 24, IBM Corp. Armonk, NY, USA), NCSS Statistical Software (2020, Kaysville Utah, USA) and GraphPad Prism (Version 6, San Diego, CA, USA).

## 3. Results and Discussions

### 3.1. Yield and G.C.–M.S. Analysis

As shown in Table 2, the most considerable quantity of essential oils extracted, calculated based on the dry weight of the plant (*v*/*w*), was recorded in L8b (5.20%, *v*/*w*), followed by L9b (3.80%, *v*/*w*), L6 (1.70%, *v*/*w*) and L7 (1.30%, *v*/*w*). It is important to note that the yield values increase at the flowering stage more than before and after flowering. As reported by several authors, the EOs concentration in inflorescences is dependent on the variety and environmental conditions [55,56,57].

In the present study, thirty-five compounds were identified with G.C.–M.S. analyses (Table 3). A percentage of 67.50 to 100% of the total EOs composition of the flowers/leaves of *L. multiflora* M were identified. It can be observed that the content (% of essential oil) of the determined compounds varies between EOs of the plants collected in the two provinces.

According to Table 4 and Figure 3, the main compound classes represented in all EOs were monoterpene hydrocarbons, oxygenated monoterpenes, sesquiterpene hydrocarbons, and oxygenated sesquiterpenes. Monoterpene hydrocarbons are the most common compounds in EOs, with a proportion ranging from 19.60% to 93.82%. L8b, L8a, L7, L4, L5 and L9b respectively are the samples that contain the most of them. Next, the monoterpenes oxygenated compounds which are present from 0.00% to 52.90% in EOs. The highest detected concentrations of these compounds are in L1/L2. They are followed by sesquiterpenes hydrocarbons present in proportions ranging from 0 to 17.60%. L1/L2 are the most concentrated in these compounds. Finally, oxygenated sesquiterpenes are the least present in the different oils. L2 and L9a are the samples that contain the most (7.90% and 5.80% respectively).

Limonene (p-mentha-1,8-diene) is the only monoterpene that appears in all EOs. It is the most predominant compound in the EOs of L4 to L8a et L9b with a percentage ranging from 25.95% to 40.70%. Moreover, it is more present in the plants sampled in M’banza Kongo (40.70%) and Cazengo (32.83%) than in Tomboco (25.95%). By comparing the chemical profiles in the table, the synthesis of Limonene showed an evident fluctuation during the different stages of plant growth. It is present throughout the phases but is slightly higher in samples taken before flowering than in those selected during and after flowering. These differences may be due to the type of harvested part of the plant, the physiological metabolism, phenology or environmental changes in the biotic and abiotic factors and the geographical aspect, which were not the same in this study [59,60,61,62].

Limonene has wide applications as a flavour additive in food, beverages, fragrances, cosmetics and household products [63]. It is one of the most common terpenes in nature and a major constituent of numerous essential oils from Citrus. [64]. In genus *Lippia*, Limonene has been identified in *Lippia turbinata* Griseb. (60.60%) [65], *Lippia alnifolia* (47.20%) [66], *Lippia alba* (Mill.) N.E. Br. ex Britton and P. Wilson (30.60 to 33.00%) [67,68], *Lippia junelliana* (Moldenke) Trunk. (23.12%) [65], and *Lippia citriodora* Kunth (syn. *Aloysia citriodora* Palau) (10.70%) [67].

According to the results (Table 3), Piperitenone would also follow Limonene in almost all EOs profiles. Among the eleven samples, it is present in eight EOs at a rate of 11.58% to 34.83%. Trans-tagetone is also detected in L4, L5, L6, L7 and L8b, with a proportion ranging from 8.50% to 12.34%. The two compounds Neral and Citral are only present in the L1 (25.3%, 21.70%) and L2 (24.40%, 28.30%) samples, respectively. We also notice the presence of p-cymene in the L1 (10.40%), Carvacrol in L9a (11.50%) and a significant concentration of Ipsenone in L4 (19.40%). Artemisia ketone has been detected in L6 (20.30%) and L9a (28.30%) and is not a common compound in the genus *Lippia*. It is most often found in different species of the genus Artemisia [69]. As for verbenone, it is present in large quantities in only L3 (55.90%). It is a natural organic compound classified as bicyclic monoterpenoids that are found naturally in a variety of plants. Verbenone is a compound that has an odour and/or flavour of camphor, celery and menthol. It can be found in many food items such as spearmint, cabbage, white cabbage, and rosemary, which makes verbenone a potential biomarker for the consumption of these food products [70]. The anti-aggregation pheromone verbenone has been used to reduce the attack rates on pines by the mountain pine beetle, *Dendroctonus ponderosae* Hopkins. In northern America, it the most damaging insect pest of lodgepole pine Pinus contorta Douglas ex Loudon [71,72]. In genus *Lippia*, it has been detected in the essential oil of *Lippia citriodora* (15.64%) and *Lippia alba* (Mill.) N.E. Brown (21.74%) [73,74].

It is interesting to note that although *Lippia* species have grown under the same edaphic and climatic conditions, there is a difference in the chemical profiles observed in plants groups from the same geographical areas. Three outgoing lots can be distinguished: L2 (Citral/Neral/Elemol), L3 (Verbenone/Ocimenone/Limonene), and L9a (Artemisia ketone/Limonene/Piperitenone). This phenotypic variability within the same species from the same region can be explained by the spatial influence of the environment on the plant [56,75]. The importance of spatial distributions in pollination, competition, herbivory, nutrient cycling and other ecological processes in the variability of the content and quality of secondary metabolites is well documented [56,57,76,77,78]. Therefore, detection and measurement of the spatial pattern are relevant for understanding phenotypic variability along a geographical gradient within the same community or the same population [79]. However, phenology and genetic or developmental stage of plant organs should also be considered [80,81]

According to previous studies, a high chemical polymorphism has also been observed at the level of African *Lippia* species. In contrast to the Angolan species, high concentrations of 1,8-cineole and Sabinene were found in oils from Nigeria [82], Togo [83], Ivory Coast [29], Benin [84] and Ghana [18]. A Geraniol, Geranial/Neral, Thymol, Linalool, Tagetone/Ipsenol, Epoxy-myrcene, p-cymene/Thymol/Ethyl acetate, Myrtenol/Linalool/1,8-cineole, and Nerolidol chemotypes were also described [29,37,82,83,85]. These studies thus demonstrate that the Angolan species is atypical in its chemical composition compared to its neighbours.

A principal component analysis (PCA) was conducted on the chemical content data for each *Lippia* oil to obtain a basic overview of the data structure and to identify similarities and specific grouping patterns. Data raw (number of principal components (PCs) and per cent variance of the first three PCs) for eight PCA models are presented in Appendix A. Mutual projections of factor scores and their loadings for the first three PCs have been presented in Figure 4, Figure 5 and Figure 6. We obtained the following variance percentage for the principal components: 26.37% (PC1), 18.70% (PC2), and 11.21% (PC3) which resulted in a model that would explain approximately 56.28% of the total variance (Figure 4). The scoring plots (Figure 5) distinguishes three oils that come out of the sample batch (L2, L8a, L9a). The loading plots (Figure 6) implies that the most influential compounds for discriminating L2 from other oils are γ-terpinene, Germacrene D, Elemol, Citral, Neral, Hedycaryol, and Zingiberene. The presence of a high concentration of Carvacrol and Artemisia Ketone separates L9a from the samples. Humulene oxide, Carvone, Trans-carveol, Trans-carvyl acetate and Caryophyllene oxide also contribute to the separation of L8a from the plants. Other oils (L1/L3/L4/L5/L6/L7/L8b/L9b), however, are joined into a single group by their common chemical composition, including Limonene, α-humulene, Ipsenone, Myrcene, p-cymene, Trans-tagetone, β-caryophyllene, Piperitenone and p-mentha-trans-2,8-dien-1-ol. We can also distinguish loading plot of Verbenone, Carveyl acetate and Vervenol in this group.

For further characterisation of the *Lippia* EOs, the hierarchical cluster analysis (HCA), followed by the heat-map using the abundance of individual volatiles were performed (Figure 7. The total HCA dendrogram (Figure 7a) showed that the volatile profile of L4/L5/L6/L7 and L3/L9b were close to each other. Also, L2, L8a, and L9a are isolated from the others by their chemical profiles. The oil distribution is the same as shown by the PCs. The HCA results generated using the abundance of the volatile compounds (Figure 7b) also showed variability between the populations from the two provinces, but also intraspecific variability between sub-groups within a population. This phenomenon has also been observed in several species of *Lippia*, such as *L. origanoides*, *L. alba*, *L. lupulina*, *L. velutina*, *L. sidoides*, *L. salviifolia*, and *L. grata* [56,86]. Furthermore, it should also be noted that for the L8/L9 plants, the EO chemical composition of their leaves (L8a/L9a) is different from that of their flowers (L8b/L9b). Indeed, the functioning of each organ can be independent of the others through authentic genetic machinery [87,88]. The chemical composition of an essential oil varies considerably not only according to individual genetic variability but also according to the phenological stage and organ of the plant.

### 3.2. Antimicrobial Vapour Phase Test

The antibacterial activity of *Lippia multiflora* Moldenke EOs has been tested using a vapour evaluation technique. It exclusively allows us to detect the antimicrobial power of EOs volatile components against three bacterial strains *Staphylococcus aureus*, *Escherichia coli* and *Pseudomonas aeruginosa*. The results are reported in Table 5.

The volatile oil of L1/L2/L3/L4/L5/L9a significantly inhibits the growth of tested microorganisms although they were more potent against *S. aureus* than against other strains of bacteria (Figure 8). Among those, L4 gave the strongest inhibition activity against all investigated pathogens with diameter values in the range of 25.67 ± 0.58 mm to 29.00 ± 1.00 mm. It seems that the presence of a high concentration of Limonene (37.20%), Ipsenone (19.40%), Piperitenone (14.00%), and Trans-tagetone (12.34%), in its oils was responsible for this higher antibacterial activity. Other essential oils L8a (17.33 ± 0.58 mm), L8b (19.33 ± 0.58 mm) and L9b (19.00 ± 1.00 mm) exhibited antimicrobial only against *P. aeruginosa*. However, L6/L7 are inefficient.

It is worthy to note from the results of the study that *P. aeruginosa* is the most resistant to the action of EOs. This bacterium is an opportunistic human pathogen responsible for several different nosocomial infections [89]. Through the presence of intrinsic resistance mechanisms and the development of irreversible mutations and adaptations, *P. aeruginosa* can survive and multiply in several environments and over a wide range of temperatures [90]. It is resistant to all types of conventional antibiotics, as it has an exceptional capacity to develop biofilms that are difficult to control [91]. The indiscriminate misuse of antimicrobial drugs would lead to an increase in bacterial resistance in chronic disease [92]. As a result, this bacterium is now a global health threat [92,93]. One approach to overcome this problem is to use essential plant oils that may represent a promising source for new resistance modifying agents [93]. In our study, most of the oils were effective against *P. aeruginosa.* However, the antibacterial activity depends on the volatility of the chemical compounds present in these EOs [94]. Indeed, the vapour generated by essential oils has a more significant antimicrobial effect than that of direct contact with these oils in liquid form. Lipophilic molecules in the aqueous phase could associate and form micelles which limit the fixation of EOs to microorganisms [95]. According to Kloucek et al., there is variability in the phenomenon of essential oils’ volatility. Each EO presents a mixture of chemical compounds that have specific volatility. Each compound diffuses into the environment with its specific speed according to its molecular weight. The vapours interact together to reach an equilibrium that would allow the destruction of microorganisms, but only in a closed environment [96].

### 3.3. Well Diffusion Agar Technique

The essential oil from the different *L. multiflora* Mold. collections were tested for their antibacterial activity against *Staphylococcus aureus*, *Escherichia coli* and *Pseudomonas aeruginosa.* Three concentrations (C^1^, C^2^, C^3^) of each EO were tested using the wells diffusion agar technique. The results in terms of inhibition zone diameter (mm) are summarised in Table 6.

In order to analyse the correlation between the chemical composition of the different essential oils and their antibacterial activity, the mean values of the diameters of the inhibition zones were subjected to a PCA and HCA analysis. First, we proceeded to the classification of essential oils according to the values of the diameters of the inhibition zones (Table 6): we have EO non-effective for a diameter equal to or less than 8 mm; effective (+) for a diameter between 8 and 14 mm; highly effective for a diameter between 14 and 20 mm and extremely effective for a diameter equal to or greater than 20 mm [97,98]. The heat-map correlation of the *Lippia* samples according to their antibacterial activity was then performed. Data raw (number of principal components (PCs) and per cent variance of the first three PCs) of PCA models are presented in Appendix A. Mutual projections of factor scores and their loadings for the first three PCs have been presented in Figure 9 and Figure 10. Starting with 11 variables, the PCA calculated three components having eigenvalues greater than one and representing together 87.97% of the total variance. PC1 accounted for 33.91%, PC2 for 30.02% and PC3 for 24.04% of the variance (Figure 11).

In the heat map (Figure 12a), almost all the essential oils of the plants used in the study showed an effect against bacteria. The best results recorded were those against *S. aureus*, followed by *P. aeruginosa* and *E. coli*, but they remain much lower than those produced by Gentamicin A (Table 6). L6C^1^/L3C^2^/L1C^3^ observed the most important effects against *S. aureus*, L1C^1^/L1C^2^/L2C^3^ against *E. coli*, and L9bC^1^/L4C^2^/L3C^3^ against *P. aeruginosa*. It is also noted that all tested concentrations of L6/L7/L8a were ineffective against *E. coli*. Similarly, for L7/L8a against *P. aeruginosa*. These samples were also recorded as ineffective by the vapour technique previously performed (Section 3.2).

The dendrogram from HCA (Figure 12b) presents two distinct main clusters, A and B. Cluster B could correspond to the L7/L8b group, in which L7C^3^/L8bC^3^ were observed to be effective (+) against *S. aureus* (11.67 ± 1.53 mm/11.00 ± 1.00 mm), while L7C^1^/C^2^/C^3^, L8bC^1^/C^2^/C^3^ showed negative results against *P. aeruginosa.* With the formation of subgroups in HCA and the results of loading and scores plots in PCA, it can be observed that oils are effectively divided according to their efficacy against the three bacteria. There are subgroups of L1/L2, which have shown better efficacy against *E. coli*, and subgroups L7/L8b, L5/L9a/L9b, L3/L4 and L6/L8a, which are relatively more effective against *S. aureus* and *P. aeruginosa*.

Two distinct groups can be observed in cluster A (Figure 12): A1 and A2. In gr. A1, we first observe subgroups (1) of L1/L2, whose members L1C^1^/L2C^1^ are effective (+) and L1C^2^/L1C^3^ highly effective (++) against *S. aureus*. There are also L1C^1^/L1C^2^, which are effective (+) against *P. aeruginosa*. and L2C^1^/L2C^3^ very effective (++) against *E.coli*. Sub-groups (2) L5/L9a/L9b, of which L5C^1^/L5C^2^/L9aC^1^/L9aC^1^ are highly effective (++) against *S. aureus*, while L5C^1^/L5C^2^/L5C^3^/L9aC^1^/C^2^/L9aC^3^ are effective (+) against *E. coli*. On the other hand, *P. aeruginosa* is sensitive (+) to L5C^3^/L9bC^3^, and highly sensitive (++) to L5C^2^/L9aC^2^/L9bC^1^ samples. In subgroup (3) of L3/L4, the L3C^1^/L3C^2^/L4C^1^/L4C^2^ are extremely effective (+++) against *S. aureus*. At the same time, L3C^1^/L3C^2^/L4C^1^/L4C^2^ are effective (+) and L3C^3^/L4C^3^ non-effective against *E. coli*. There are also L3C^1^/L4C^1^, which are effective (+) and L3C^2^/L4C^2^ highly effective (++) against *P. aeruginosa*. Finally, gr. A2 of L6/L8a showed negative results against *E coli* (L6C^1^/C^2^/C^3^, L8aC^1^/C^2^/C^3^), but they are highly effective (++) against *P. aeruginosa* (L6C^3^/L8C^3^*).*

According to our study, the activity of the EOs against gram-negative bacteria was greater than that against gram-positive bacteria. The results were not in line with previous research. Indeed, gram-negative bacteria are more resistant to essential oils than Gram-positive bacteria. Their outer membrane has a thick layer of impermeable peptidoglycan. It makes it difficult for antimicrobial agents to pass through and confers rigidity and resistance to gram-negative bacteria [99,100,101]. The variability of the antibacterial activity of the different *Lippia* collections is probably due to the synergistic effect that could occur in the oil and thus potentiate or not its biological activity. According to several studies, the antibacterial activity depends on the quantity of the chemical constituent in the EOs, i.e., at low concentrations they can interfere with the enzymes involved in energy production and at higher concentrations they can denature proteins [102,103].

In our study, the plants that have shown the highest activity against the three bacteria strains are the ones that contain more Limonene. This one is reported to have antimicrobial properties against a wide range of bacteria [104,105,106,107,108] related to its chemical nature. Indeed, Limonene belongs to the family of cyclic monoterpene hydrocarbons known for their effectiveness against bacteria. Their mechanism of action consists of an accumulation in the plasma membrane and dissipation of the protons motive force causing a loss of membrane integrity [109,110]. On the other hand, the results of our study also allow us to suggest that the antibacterial activity of the different *Lippia* samples could not be attributed solely to Limonene. Common compounds such as Piperitenone, Ketone artemisia Ipsenone, Ocimenone, Citral, Neral, Elemol and other minor compounds have been detected. They could play a critical role by producing a synergistic and additives effect. Their antimicrobial activity is not attributable to a specific mechanism, but there are several targets in the cell [101].

In the last part of this study, we were able to determine the Minimum Inhibitory Concentration (MIC) and Minimum Bactericidal Concentration (MBC) values following standard guidelines published by the National Committee for Clinical Laboratory Standards to characterise antibiotic activity [111]. The MIC/MBC ratio was then calculated to determine the antibacterial power of the EOs studied. The interpretation of the results was based on the classification of Shanmughapriya et al. (2010). EO is considered to be bactericidal when the ratio is less than or equal to 2 and bacteriostatic when it is greater than 2 [112]. The results are summarised in Table 7.

There are bacteriostatic (ration greater or equal to 2), bactericidal (whose ratio is between 1 and 2) and very bactericidal oils (whose ration is equal or less than 1) according to the calculated ratios. In the results of activity against *S. aureus*, different levels of activity were found with MIC values ranging from 42.52 ± 20.05 to 113.38 μg/mL and MBC values ranging from 47.24 ± 13.36 μg/mL to 170.07 ± 80.16μg/mL (see Table 7). L7 is the only oil that is bacteriostatic against bacteria. L1/L2/L6/L8a/L8b/L9b EOs are bactericidal, whereas L3/L4/L5/L9a are very bactericidal. For *E. coli*, the MIC values are from 85.04 ± 40.09 to 113.38 ± 0.00 μg/mL, and the MBC values are ranging from 85.04 ± 40.09 to 226.75 ± 0.00 μg/mL. The L6/L7/L8a are bacteriostatic, L3/L4/L6/L7/L8a bactericidal, and L1L2/L5/L8b/L9a/L9b very bactericidal EOs. The MIC values for *P. aeruginsa* are between 42.52 ± 20.05 and 113.38 ± 0.00 μg/mL, and those for MBC are in the range 56.69 ± 0.00 and 226.75 ± 0.00μg/mL. The oils that have shown bacteriostatic effects are L4 and L7. The others, L5/L8b/L9a are bactericidal and L1/L2/L3/L6/L8a/L9b very bactericidal EOs.

Due to the wide variation in the chemical profiles of *L. multiflora* Mold. essential oils, the antibacterial activity is also well diversified in the literature [10,32,113,114,115]. According to the study by Bassolé et al. (2010), *Lippia* from Burkina-Faso was the most effective among the plants tested against *S. aureus* with a MIC = 1.2 ± 0 mg/mL [32]. The Ivory Coast plant gave better antibacterial activity against *S. aureus* and *P. aeruginosa* with MIC = MBC = 0.9 mg/mL [32,114]. Moreover, MIC values range from 3 to 96 × 10^−3^mL^−1^ for *Lippia* from Gabon and no antibacterial activity was observed for Nigerian plant against *S. aureus* or *E. coli* [82,116]. The antimicrobial activity of any essential oil is not attributable to just one mechanism but to several ones widely described in the literature [89]. Factors such as climatic and environmental conditions, the origin of the plant, the plant’s adaptive metabolism, the harvesting season, the part of the plant involved in the extraction, the distillation conditions, the microbiological methods used, as well as the susceptibility of the bacterial strains would make it difficult to compare the results obtained by different groups of researchers [117,118].

## 4. Conclusions

The composition and antibacterial effects of *Lippia multiflora* Moldenke from two regions of Angola were assessed. Samples were collected during, before and after flowering. The highest yield of essential oils among the 11 samples was obtained from flowers (5.2%) and leaves (3.8%, 1.7% and 1.3%). They were harvested in the municipality of Tombocco, province of Zaire. During this study, we observed that the composition of EOs also changes depending on the part of the plant. The main components identified were monoterpene hydrocarbons which reached proportions ranging from 19.60% to 93.82% of all compounds identified by GC-MS. The highest accumulation was Limonene (from 25.95% to 40.70%) in *Lippia* samples collected before flowering. Piperitenone, Trans-tagetone, Neral, Citral, Elemol, p-cymene, Myrcene, Carvacrol, and Ipsenone were also identified, and their proportion varied between-sample. Two new compounds, including Verbenone and Artemisia ketone, were detected for the first time in the essential oil of *Lippia multiflora* Moldenke from Angola. The analysis of *Lippia*’s EOs by principal component analysis (PCA) and hierarchical cluster analysis (HCA), followed by the highlighting of the heat map of the volatile compounds’ abundance allowed us to identify a variability between the populations from the two provinces, but also intraspecific variability between sub-groups within a population. Due to this great chemical variability, the antibacterial activity is also well-diversified. Thanks to HCA and PCA, we were able to discern the similarities between subgroup of chemotypes and differentiate their respective antibacterial activity. According to MIC and MBC values, *Lippia* essential oils showed remarkable activity against the three bacteria *S. aureus* and *E. coli*, and *P. aeruginosa*. These results suggest that Angolan *Lippia multiflora* Moldenke can be used as a safe alternative source in the pharmaceutical, cosmetic and food industries.

## Figures and Tables

**Figure 1 molecules-26-00155-f001:**
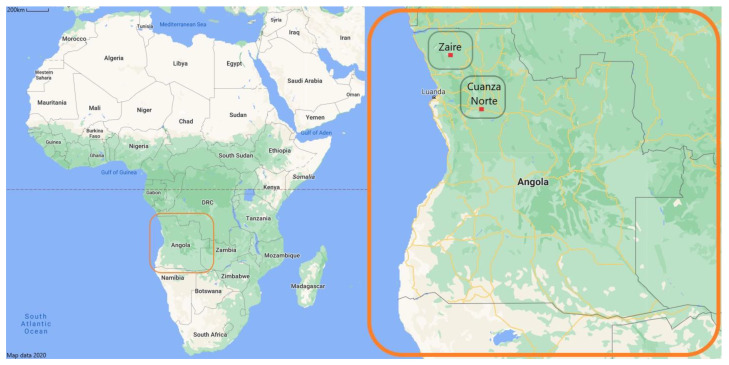
The topography of Angola, indicating provincial boundaries and regions of study [49].

**Figure 2 molecules-26-00155-f002:**
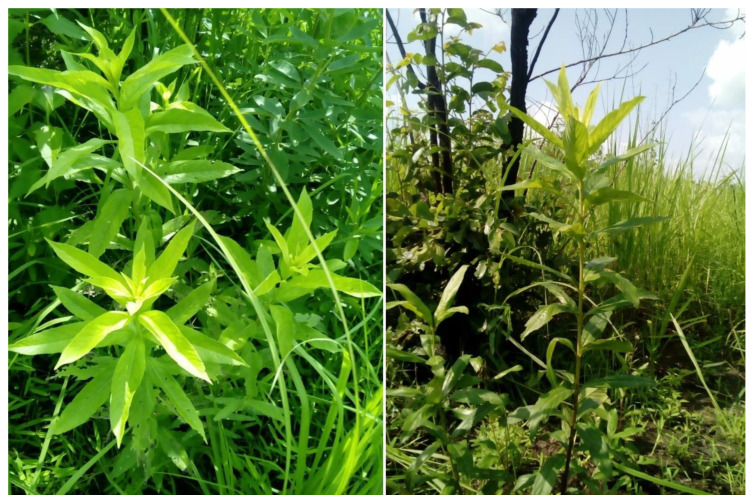
Angolan *Lippia multiflora* Moldenke.

**Figure 3 molecules-26-00155-f003:**
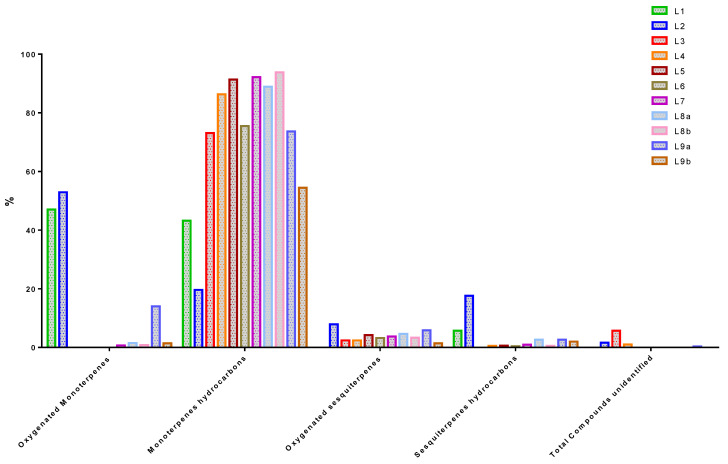
Representation of the main chemical compound classes identified in the essential oils (Eos) of *Lippia multiflora* Moldenke (L1/L2/L3/L4/L5/L6/L7/L8a/L8b/L9a/L9b).

**Figure 4 molecules-26-00155-f004:**
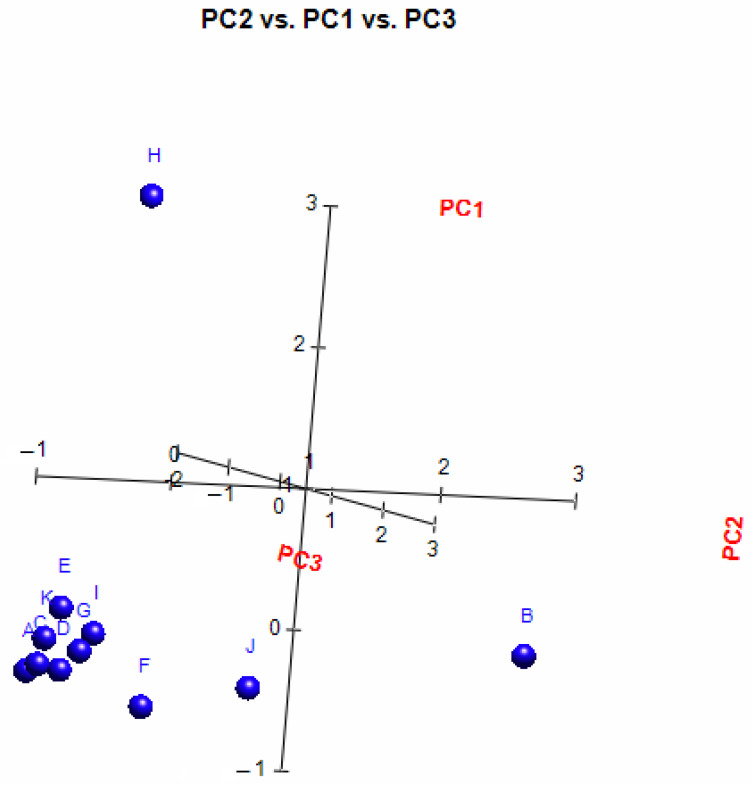
PCs scatter plot of Angolan *Lippia multiflora* Moldenke essential oils. The x, y, and z axes exhibit the distribution of each oil according to their chemical composition. A = L1; B = L2; C = L3; D = L4; E = L5; F = L6; G = L7; H = L8a; I = L8b; J = L9a; K = L9b.

**Figure 5 molecules-26-00155-f005:**
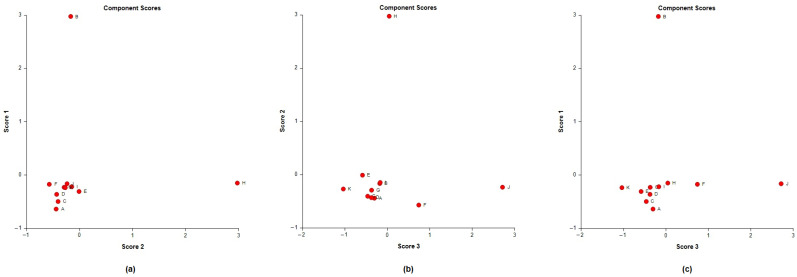
Principal component analysis (Score plots) of the essential oils obtained from Angolan *Lippia multiflora* Moldenke. (**a**) Score plot of PC1 vs. PC2; (**b**) Score plots of PC2 vs. PC3; (**c**) Score plots of PC1 vs. PC3. A = L1; B = L2; C = L3; D =L4; E = L5; F = L6; G = L7; H = L8a; I = L8b; J = L9a; K = L9b.

**Figure 6 molecules-26-00155-f006:**
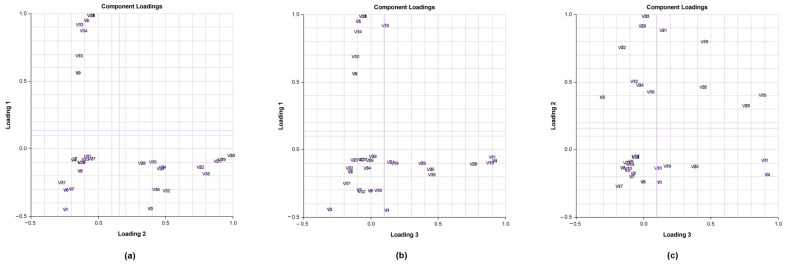
Principal component analysis (Loading plots) of the constituents of the essential oil obtained from Angolan *Lippia multiflora* Moldenke. (**a**) Loading of PC1 vs. PC2; (**b**) Loading of PC3 vs. PC1; (**c**) Loading of PC2 vs. PC3. V1 = Myrcene; V2 = p-cymene; V3 = Limonene; V4 = Artemisia ketone; V5 = γ-terpinene; V6 = Ipsenone; V7 = Trans-tagetone; V8 = Verbenone; V9 = Neral; V10 = Citral; V11 = Carvacrol; V12 = Piperitenone; V13 = Germacrene D; V14 = Elemol; V15 = β-farnesene; V16 = β-caryophyllene; V17 = α-humulene; V18 = Caryophyllene oxide; V19 = Thymol acetate; V20 = Thymol; V21 = Trans-carveol; V22 = Trans carvyl acetate; V23 = Vervenol; V24 = Guaiol; V25 = δ-cadinol; V26 = Isopiperitenone; V27 = Carveyl acetate; V28 = α-pinene; V29 = Carvone; V30 = Geraniol; V31 = Zingiberene; V32 = Hedycaryol; V33 = Humulene oxide; V34 = p-mentha-trans-2,8-dien-1-ol; V35 = Linalool.

**Figure 7 molecules-26-00155-f007:**
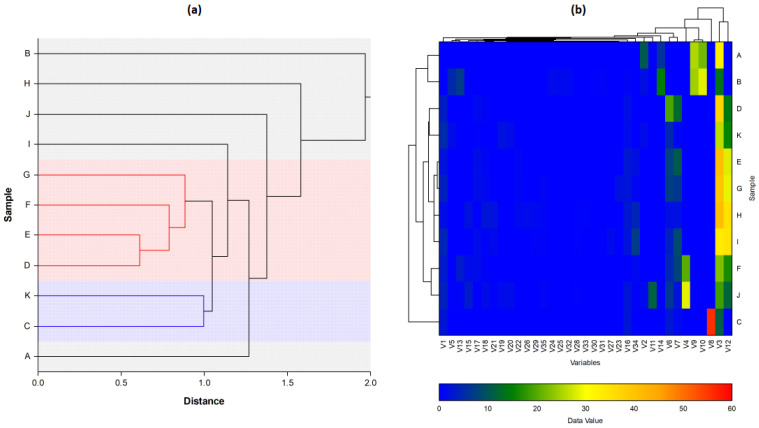
Hierarchical cluster analysis (**a**) and heat-map using the abundance (**b**) of all volatiles compounds of different Angolan *Lippia multiflora* Moldenke EOs (n = 11). A = L1; B = L2; C = L3; D = L4; E = L5; F = L6; G = L7; H = L8a; I = L8b; J = L9a; K *= L9b*. V1 = Myrcene; V2 = p-cymene; V3 = Limonene; V4 = Artemisia ketone; V5 = γ-terpinene; V6 = Ipsenone; V7 = Trans-tagetone; V8 = Verbenone; V9 = Neral; V10 = Citral; V11 = Carvacrol; V12 = Piperitenone; V13 = Germacrene D; V14 = Elemol; V15 = β-farnesene; V16 = β-caryophyllene; V17 = α-humulene; V18 = Caryophyllene oxide; V19 = Thymol acetate; V20 = Thymol; V21 = Trans-carveol; V22 = Trans carvyl acetate; V23 = Vervenol; V24 = Guaiol; V25 = δ-cadinol; V26 = Isopiperitenone; V27 = Carveyl acetate; V28 = α-pinene; V29 = Carvone; V30 = Geraniol; V31 = Zingiberene; V32 = Hedycaryol; V33 = Humulene oxide; V34 = p-mentha-trans-2,8-dien-1-ol; V35 = Linalool.

**Figure 8 molecules-26-00155-f008:**
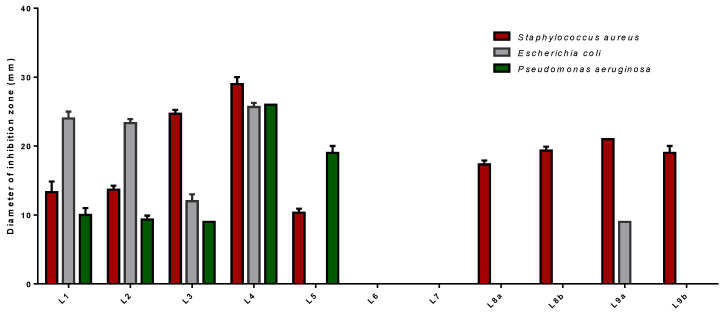
Antibacterial activity of *Lippia multiflora* Moldenke essential oils by vapour phase test.

**Figure 9 molecules-26-00155-f009:**
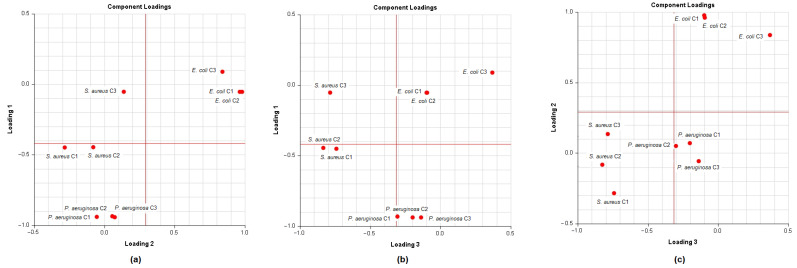
Principal component analysis (Loading plots) of antibacterial activity of Angolan *Lippia multiflora* Moldenke EOs. The wells diffusion agar technique tested three concentrations (C^1^, C^2^, C^3^) against *Streptococcus aureus*, *Pseudomonas aeruginosa* and *Escherichia coli*. (**a**) Loading plost of PC1 vs. PC2; (**b**) Loading plots of PC1 vs. PC3; (**c**) Loading plots of PC2 vs. PC3.

**Figure 10 molecules-26-00155-f010:**
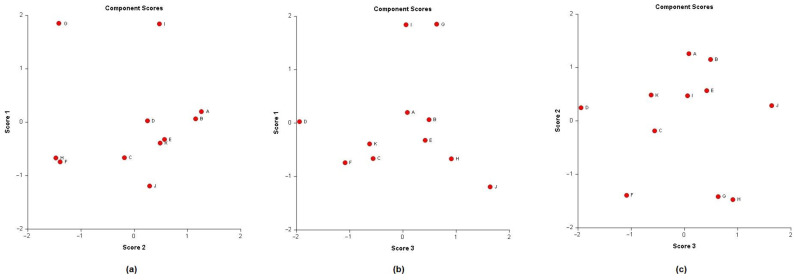
Principal component analysis (Score plots) of antibacterial activity of Angolan *Lippia multiflora* Moldenke EOs. (**a**) Score plot of PC1 vs. PC2; (**b**) Score plots of PC1 vs. PC3; (**c**) Score plots of PC2 vs. PC3. A = L1; B = L2; C = L3; D = L4; E = L5; F = L6; G = L7; H = L8a; I = L8b; J = L9a; K = L9b.

**Figure 11 molecules-26-00155-f011:**
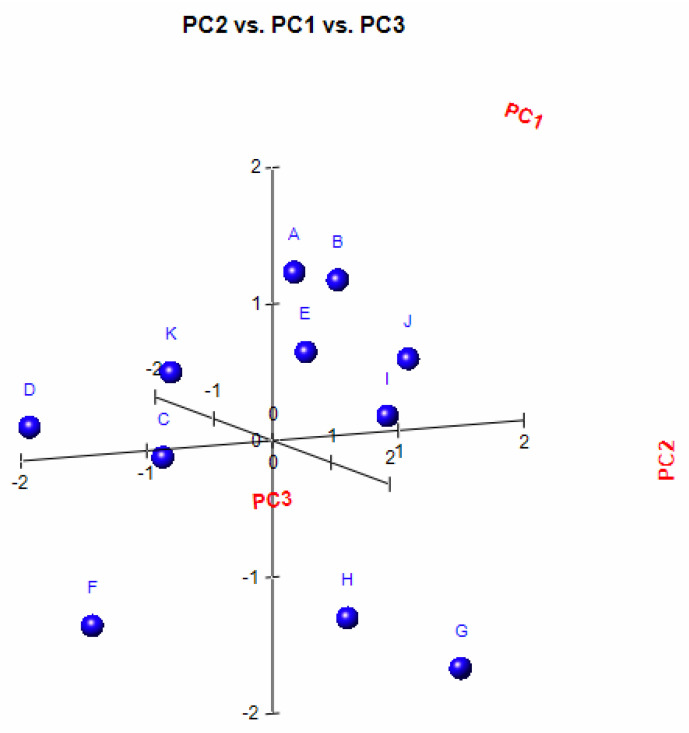
PCs scatter plot of Angolan *Lippia multiflora* Moldenke essential oils. The x, y, and z axes exhibit the distribution of each oil according to their antibacterial activity against *Streptococcus aureus*, *Pseudomonas aeruginosa* and *Escherichia coli.* A = L1; B = L2; C = L3; D = L4; E = L5; F = L6; G = L7; H = L8a; I = L8b; J = L9a; K = L9b.

**Figure 12 molecules-26-00155-f012:**
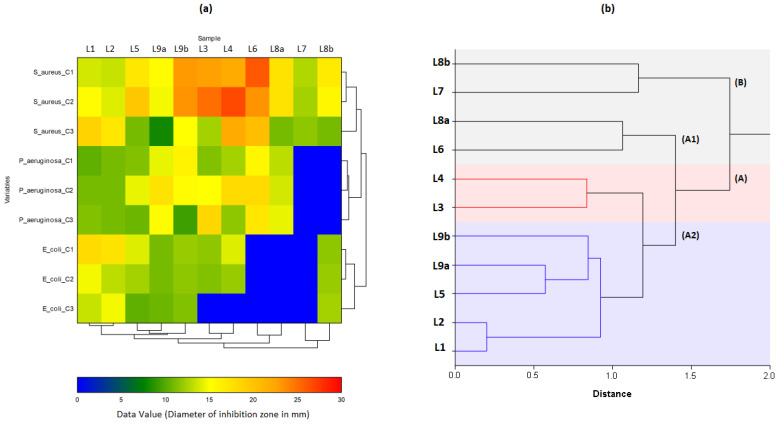
Hierarchical cluster analysis (**a**) and heat-map using diameter of antibacterial inhibition zone (mm) (**b**) of all Angolan *Lippia multiflora* Moldenke EOs (*n* = 11).

**Table 1 molecules-26-00155-t001:** Harvesting areas of the different parts of the plant *Lippia multiflora* Moldenke.

Sample	Part of Plant	Harvesting Time	Province	Municipalities
L1	Leaves	F	Cuanza Norte	Cazengo
L2	Leaves	F		
L3	Leaves	AF	Zaire	M’banza Kongo
L4	Leaves	BF		
L5	Leaves	BF		
L6	Leaves	AF		Tomboco
L7	Leaves	BF		
L8a/b	Leaves/flowers	BF/F		
L9a/b	Leaves/flowers	BF/F		

BF = before flowering period; F = during the flowering period; AF = after flowering period.

**Table 2 molecules-26-00155-t002:** Yields of essential oils of *Lippia multiflora* Moldenke sampled.

Sample	Part of Plant	Yield (%) *v*/*w*	Sample	Part of Plant	Yield (%) *v*/*w*
L1	Leaves	0.80	L6	Leaves	1.70
L2	Leaves	0.90	L7	Leaves	1.30
L3	Leaves	1.00	L8a/b	Leaves/flowers	1.20/5.20
L4	Leaves	0.40	L9a/b	Leaves/flowers	1.10/3.80
L5	Leaves	1.00			

**Table 3 molecules-26-00155-t003:** Chemical composition (content %) of eleven *L. multiflora* Mold. EOs.

RRI ^1^	Compounds	L1	L2	L3	L4	L5	L6	L7	L8a	L8b	L9a	L9b
939	α-pinene	0.00	0.00	Tr	Tr	Tr	Tr	Tr	0.18	0.22	0.37	Tr
991	Myrcene	0.00	0.00	2.40	3.40	2.10	3.00	6.80	1.40	4.30	3.90	4.90
1026	p-cymene	10.40	1.60	0.00	0.00	0.00	0.00	0.00	0.00	0.00	0.90	1.20
1031	Limonene	32.80	13.30	11.20	37.20	40.60	22.60	39.60	40.70	33.30	18.70	26.00
1062	Artemisia ketone	0.00	0.00	0.00	0.00	0.00	20.30	0.00	0.00	0.00	28.30	0.00
1062	γ-terpinene	0.00	4.70	0.00	0.00	0.00	0.00	0.00	0.00	0.00	0.37	1.45
1083	Ipsenone	0.00	0.00	2.70	19.40	7.60	4.50	7.30	1.60	3.80	6.50	4.70
1093	Linalool	0.00	Tr	0.00	0.00	0.00	0.00	0.70	0.60	0.40	0.60	Tr
1122	p-mentha-trans-2,8-dien-1-ol	0.00	0.00	0.00	0.00	1.80	0.80	0.70	4.30	7.10	0.40	Tr
1146	(E)-tagetone (trans-tagetone)	0.00	0.00	0.90	12.34	10.53	7.89	6.56	1.70	8.50	2.20	0.70
1150	Vervenol	0.00	0.00	0.00	0.00	0.00	0.00	1.90	0.00	0.00	0.00	0.00
1189	α-terpineol	0.00	0.00	Tr	0.00	0.00	Tr	0.00	0.00	Tr	0.60	0.00
1204	Verbenone	0.00	0.00	55.90	0.00	0.00	0.00	0.00	0.00	0.00	0.00	0.00
1217	Trans–carveol	0.00	0.00	0.00	0.00	0.00	0.00	0.00	2.10	1.10	0.40	Tr
1228	Neral	25.30	24.40	0.00	0.00	0.00	0.00	0.00	0.00	0.00	0.00	0.00
1240	Citral	21.70	28.30	0.00	0.00	0.00	0.00	0.00	0.00	0.00	0.00	0.00
1242	Carvone	0.00	0.00	0.00	0.00	0.00	0.00	Tr	0.90	0.40	0.00	0.00
1255	Geraniol	0.00	0.20	0.00	0.00	0.00	0.00	0.00	0.00	0.00	0.00	0.00
1277	Trans-carvyl acetate	0.00	0.00	Tr	0.00	0.69	0.00	0.60	1.00	0.00	0.00	0.00
1290	Thymol	0.00	0.00	0.00	0.00	0.00	0.00	0.00	0.00	0.00	1.30	1.40
1299	Carvacrol	0.00	0.00	0.00	0.00	0.00	0.00	0.00	0.00	0.00	11.50	0.00
1342	Piperitenone	0.00	0.00	Tr	14.00	28.00	16.40	28.70	34.80	34.30	11.60	15.50
1355	Thymol Acetate	0.00	0.00	0.00	0.00	0.00	0.00	0.00	0.00	0.00	1.20	2.00
1418	β-caryophyllene	0.00	0.50	2.40	1.50	2.70	0.00	2.30	2.70	2.10	2.30	1.20
1453	α-humulene	0.00	0.00	Tr	0.90	1.50	1.30	1.40	0.00	1.20	0.00	0.20
1458	β-farnesene	0.00	0.00	Tr	0.00	0.00	1.30	0.00	1.90	0.00	3.00	Tr
1480	Germacrene D	0.00	6.90	Tr	Tr	Tr	0.60	0.00	0.00	Tr	0.50	Tr
1495	Zingiberene	0.00	0.50	0.00	0.00	0.00	0.00	0.00	0.00	0.00	0.00	0.00
1530	Hedycaryol	0.00	0.80	0.00	0.00	0.00	0.00	0.00	0.00	0.00	0.00	0.00
1547	Elemol	5.70	14.60	0.00	0.00	0.00	0.00	0.00	0.00	0.00	0.00	0.00
1581	Caryophyllene oxide	Tr	0.00	Tr	0.50	0.60	0.40	0.90	2.30	0.50	1.40	Tr
1595	Guaiol	0.00	1.20	0.00	0.00	0.00	0.00	0.00	0.00	0.00	0.00	0.00
1606	Humulene oxide	0.00	0.00	0.00	0.00	0.00	0.00	0.00	0.30	0.00	Tr	0.00
1636	δ-cadinol	0.00	1.00	0.00	0.00	0.00	0.00	0.00	0.00	0.00	0.00	0.00
1832	Isopiperitenone	0.00	0.00	0.00	0.00	0.00	0.00	0.00	1.14	0.00	0.00	0.00
1321.9	Carveyl acetate	0.00	0.00	0.00	0.00	0.00	0.00	0.00	0.00	1.20	0.00	0.00
	% total compounds identified	95.90	98.00	75.50	89.24	96.12	79.09	97.46	97.62	98.42	96.04	59.25
	% total compounds unidentified	Tr	1.60	5.70	1.00	0.00	0.00	0.00	0.00	0.00	0.30	0.00
	Mixture	Tr	0.00	0.00	0.00	0.00	0.65	0.00	2.42	0.00	1.40	Tr

^1^ RRI: Relative retention index [18,58]. Tr: Compounds present in traces; whose percentage is less than 0.10%. Mixture: it is a set of two or more compounds whose peaks cannot be separated and facilitate their identification.

**Table 4 molecules-26-00155-t004:** Proportions of the main chemical compound classes identified in the EOs of *Lippia multiflora* Moldenke.

	Sample										
**Compound Classes (%)**	L1	L2	L3	L4	L5	L6	L7	L8a	L8b	L9a	L9b
**Oxygenated Monoterpenes**	47.00	52.90	0.00	0.00	0.00	0.00	0.70	1.50	0.80	14.00	1.40
**Monoterpenes Hydrocarbons**	43.20	19.60	73.10	86.34	91.32	75.49	92.16	88.92	93.82	73.64	54.45
**Oxygenated Sesquiterpenes**	0.00	7.90	2.40	2.40	4.20	3.20	3.70	4.60	3.30	5.80	1.40
**Sesquiterpenes Hydrocarbons**	5.70	17.60	0.00	0.50	0.60	0.40	0.90	2.60	0.50	2.60	2.00
**Total Compounds Unidentified**	0.00	1.60	5.70	1.00	0.00	0.00	0.00	0.00	0.00	0.30	0.00

**Table 5 molecules-26-00155-t005:** Inhibition zone diameter (values are means ± SD in mm) of essential oil against *Staphylococcus aureus*, *Escherichia coli* and *Pseudomonas aeruginosa.*

Sample	Inhibition Zone Diameter (mm)		
	Tested Microorganisms (Strains)		
	*S. aureus* ^a^	*E. coli* ^b^	*P. aeruginosa* ^c^
L1	13.33 ± 1.53	24.00 ± 1.00	10.00 ± 1.00
L2	13.67 ± 0.58	23.33 ± 0.58	9.33 ± 0.58
L3	24.67 ± 0.58	12.00 ± 1.00	9.00 ± 0.00
L4	29.00 ± 1.00	25.67 ± 0.58	26.00 ± 0.00
L5	10.33 ± 0.58	0.00 ± 0.00	19.00 ± 1.00
L6	0.00 ± 0.00	0.00 ± 0.00	0.00 ± 0.00
L7	0.00 ± 0.00	0.00 ± 0.00	0.00 ± 0.00
L8a	17.33 ± 0.58	0.00 ± 0.00	0.00 ± 0.00
L8b	19.33 ± 0.58	0.00 ± 0.00	0.00 ± 0.00
L9a	21.00 ± 0.00	9.00 ± 0.00	0.00 ± 0.00
L9b	19.00 ± 1.00	0.00 ± 0.00	0.00 ± 0.00
DMSO 5%	0.00 ± 0.00	0.00 ± 0.00	0.00 ± 0.00

^a, b, c^ All Values differ at *p* < 0.0001.

**Table 6 molecules-26-00155-t006:** The diameter of the inhibition of *Lippia multiflora* Moldenke EOs (Values are means ± SD in mm; all values differs at *p* < 0.05).

	Tested Microorganisms (Strains)								
	*S. aureus*			*E. coli*			*P. aeruginosa*		
	Concentrations *								
Sample	C^1^	C^2^	C^3^	C^1^	C^2^	C^3^	C^1^	C^2^	C^3^
L1	13.66 ± 2.08	15.33 ± 1.53	18.66 ± 0.57	18.00 ± 1.00	14.67 ± 1.15	13.33 ± 1.53	10.33 ± 0.58	11.00 ± 1.00	11.33 ± 1.15
L2	13.33 ± 1.52	14.00 ± 1.00	17.00 ± 1.00	17.33 ± 2.08	13.00 ± 2.65	14.67 ± 1.15	11.00 ± 1.00	11.00 ± 1.00	11.00 ± 1.00
L3	22.66 ± 2.51	25.00 ± 1.00	12.33 ± 1.53	11.67 ± 0.58	11.33 ± 0.58	0.00 ± 0.00	11.33 ± 0.58	15.00 ± 1.00	18.33 ± 1.53
L4	22.00 ± 1.00	26.67 ± 0.58	22.00 ± 5.29	14.00 ± 1.00	12.00 ± 1.00	0.00 ± 0.00	12.33 ± 0.58	18.00 ± 1.00	11.68 ± 0.58
L5	17.00 ± 1.00	19.67 ± 0.58	11.00 ± 1.00	14.00 ± 1.00	12.33 ± 1.15	10.33 ± 0.58	11.33 ± 0.58	14.33 ± 0.58	10.67 ± 1.53
L6	26.00 ± 1.00	23.33 ± 1.53	20.33 ± 1.53	0.00 ± 0.00	0.00 ± 0.00	0.00 ± 0.00	15.67 ± 0.58	18.00 ± 3.00	17.00 ± 1.00
L7	12.67 ± 1.15	12.33 ± 0.58	11.67 ± 1.53	0.00 ± 0.00	0.00 ± 0.00	0.00 ± 0.00	0.00 ± 0.00	0.00 ± 0.00	0.00 ± 0.00
L8a	17.33 ± 1.15	17.33 ± 1.15	11.00 ± 1.00	0.00 ± 0.00	0.00 ± 0.00	0.00 ± 0.00	13.00 ± 1.00	13.67 ± 1.53	14.33 ± 2.08
L8b	16.67 ± 1.15	15.67 ± 1.53	11.00 ± 1.00	11.67 ± 1.15	12.00 ± 1.00	12.33 ± 1.15	0.00 ± 0.00	0.00 ± 0.00	0.00 ± 0.00
L9a	15.33 ± 0.58	14.67 ± 0.58	8.00 ± 6.93	11.00 ± 1.00	11.00 ± 1.00	10.67 ± 1.15	14.33 ± 0.58	17.33 ± 2.08	15.33 ± 0.58
L9b	23.00 ± 1.00	23.33 ± 2.52	15.00 ± 3.00	12.00 ± 1.00	11.67 ± 0.58	11.33 ± 0.58	16.00 ± 1.73	15.33 ± 0.58	9.33 ± 2.31
Control Compound A	47.00 ± 0.00			0.00 ± 0.00			0.00 ± 0.00		
Control Compound B	45.00 ± 0.00			42.00 ± 0.00			50.00 ± 0.00		
Control Compound C	0.00 ± 0.00			0.00 ± 0.00			0.00 ± 0.00		

* C^1^ = 63.49 mg/well, C^2^ = 31.75 mg/well, C^3^ = 6.35 mg/well. Control compound A: Penicillin = 0.05 mg/mL. Control compound B: Gentamicin A = 10 mg/mL. Control compound C: MHB broth (+5% DMSO).

**Table 7 molecules-26-00155-t007:** Minimum Inhibitory Concentration (MIC) and Minimum Bactericidal Concentration (MBC) of *L. multiflora* Mold. essential oils.

	Tested Microorganisms (Strains)					
	*S. aureus*		*E. coli*		*P. aeruginosa*	
	MIC/MBC (µg/mL)					
Samples	MIC	MBC	MIC	MBC	MIC	MBC
L1	113.38 ± 0.00	170.07 ± 80.16	85.04 ± 40.09	85.04 ± 40.09	113.38 ± 0.00	113.38 ± 0.00
L2	113.38 ± 0.00	170.07 ± 80.16	85.04 ± 40.09	85.04 ± 40.09	113.38 ± 0.00	113.38 ± 0.00
L3	113.38 ± 0.00	113.38 ± 0.00	113.38 ± 0.00	170.07 ± 80.16	56.69 ± 0.00	56.69 ± 0.00
L4	85.04 ± 40.09	85.04 ± 40.09	113.38 ± 0.00	170.07 ± 80.16	113.38 ± 0.00	170.07 ± 80.16
L5	113.38 ± 0.00	113.38 ± 0.00	113.38 ± 0.00	113.38 ± 0.00	113.38 ± 0.00	170.07 ± 80.16
L6	42.52 ± 20.05	75.59 ± 53.45	113.38 ± 0.00	226.75 ± 0.00	42.52 ± 20.05	75.59 ± 53.45
L7	56.69 ± 0.00	113.38 ± 0.00	113.38 ± 0.00	226.75 ± 0.00	113.38 ± 0.00	56.69 ± 0.00
L8a	56.69 ± 0.00	85.04 ± 40.09	113.38 ± 0.00	226.75 ± 0.00	113.38 ± 0.00	113.38 ± 0.00
L8b	56.69 ± 0.00	85.04 ± 40.09	113.38 ± 0.00	113.38 ± 0.00	113.38 ± 0.00	226.75 ± 0.00
L9a	56.69 ± 0.00	56.69 ± 0.00	113.38 ± 0.00	113.38 ± 0.00	42.52 ± 20.05	75.59 ± 53.45
L9b	42.52 ± 20.05	47.24 ± 13.36	113.38 ± 0.00	113.38 ± 0.00	56.69 ± 0.00	56.69 ± 0.00
MHB Broth +5 % DMSO	0.00 ± 0.00	0.00 ± 0.00	0.00 ± 0.00	0.00 ± 0.00	0.00 ± 0.00	0.00 ± 0.00

## Data Availability

Plants of *Lippia multiflora* Moldenke were used in this study. All plant parts were identified by Samba, N. and Nelo, M. (University of Kimpa Vita, N’dalatando, Province of Cuanza Norte, Angola); Christin, H.; Mateus, D.F.D; Christoph, N. and Thea, L. (Faculty of Biology, Institute of Botany, Technische Universität Dresden, 01062, Dresden, Germany) in “Economic Potential of Selected Native Plants from Cuanza Norte, Northern Angola Mix study”. Collection and export permits were obtained from the Ministry of Environment Angola and the Province Government of Cuanza Norte and Zaire.

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
