# Peer review of "Chemical Composition and Antibacterial Activity of Lippia multiflora Moldenke Essential Oil from Different Regions of Angola"

_molecules, 2020, doi:10.3390/molecules26010155_

Round 1

Reviewer 1 Report

Ths present paper described the chemical composition and antibacterial activity of Lippia multiflora Moldenke essential oils (EOs) collected in different regions of Angola. Analysis of the oils by GC/MS identified thirty-five components representing 67.5 to 100% of the total oils. Monoterpene hydrocarbons were the most prevalent compounds, followed by oxygenated monoterpenes. In the vapour phase test, a single oil was the most effective against all the pathogens studied. The principal component analysis (PCA) and hierarchical cluster analysis (HCA) of components of the selected EOs and inhibition zone diameter values of agar wells technique allowed the identification of two groups and subgroups of species. Each group of essential oils constituted a chemotype responsible for their bacterial inhibition capacity. In my opinion, this is an interesting and well-conducted study - therefore it must be accepted for publication in Molecules after minor revision as follow:

  1. Substitute --- by 0.00 in Table 1.
  2. The percentages must be presented using the same unities
  3.  Line 183 - "% of oil" - what it means?
  4. line 209-214 and other parts of this manuscript - Lippia must be presented in italic. Please, revise it.
  5. Table 4 - what was the positive control used in the assays?
  6.  The conclusion is too poor - please revise it.

Reviewer 2 Report

Please improve the tables layout.

Please to modify the figure and table captions, trying to apply always the same layout. For example, in figure 2 the caption is central and in figure 3 not.

Figure 1. The images are blurry.

Lines 67-70. To check the grammar English.

Lines 90-95. to check the grammar English.

Table 1. To add a column for the harvesting periods.

Line 100. Water-distillation is a hydro-distillation.

Line 101. Is not specified the distillation time.

Line 174. L8. Which one? L8a or L8b?

Table 3. Dots, not commas for numbers!

To define the range in which Tr is referred on.

To define what are Total unidentified, Mixture, %of oil.

For the L1 sample the reported % total identified is 99.4, but if we calculate the sum of the % content showed in the table (10.4+32.8+25.3+21.7+5.7) the result is 95.9. The difference is 3.5.

For the L2 sample the reported % total identified is 98.2, but if we calculate the sum of the % content showed in the table (1.6+13.3+4.7+24.4+28.3+0.2+0.5+6.9+0.5+0.8+14.6+1.2+1.09) the result is 98. The difference is 0.2.

For the L3 sample the reported % total identified is 94.3, but if we calculate the sum of the % content showed in the table (2.4+11.2+2.7+0.9+55.9+2.4) the result is 75.5. The difference is 18.8.

For the L4 sample the reported % total identified is 99, but if we calculate the sum of the % content showed in the table (3.4+37.2+19.4+9.9+14.0+1.5+0.9+0.5) the result is 86.8. The difference is 12.2.

For the L5 sample the reported % total identified is 100, but if we calculate the sum of the % content showed in the table (2.1+40.6+7.6+1.8+2.2+0.69+28.0+2.7+1.5+0.6) the result is 87.78. The difference is 12.21.

For the L6 sample the reported % total identified is 96.7, but if we calculate the sum of the % content showed in the table (3.0+22.6+20.3+4.5+0.8+2.8+16.4+1.3+1.3+0.6+0.4+) the result is 74. The difference is 22.7.

For the L7 sample the reported % total identified is 100, but if we calculate the sum of the % content showed in the table (6.8+39.6+7.3+0.7+0.7+2.5+1.9+0.6+28.7+2.3+1.4+0.9) the result is 93.4. The difference is 6.3.

L8a is ok.

For the L8b sample the reported % total identified is 99, but if we calculate the sum of the % content showed in the table (0.22+4.3+33.3+3.8+0.4+7.1+8.5+1.1+0.4+34.3+2.1+.2+0.5) the result is 98.42. The difference is 0.58.

For the L9a sample the reported % total identified is 67.5, but if we calculate the sum of the % content showed in the table (0.37+3.9+0.9+18.7+28.3+0.37+6.5+0.6+0.4+2.2+0.6+0.4+1.3+11.5+11.6+1.2+2.3+3.0+0.5+1.4) the result is 59.25. The difference is 8.25.

Why are there these differences? Neither Traces are not adequate to explain them, because often the difference value is too much wide, and neither the total unidentified values are not enough to elucidate them.

The same decimal number in the text and in the table 3 are advisable.

Line 211. Limonene…is one of the most abundant terpenes in cannabis (16% of the essential oils fraction). To insert reference.

Lines 211-214. To separate the reference for each plant

215-216. please to review the period.

Line 217. Transgetone is not showed in the table and is not possible to find it in scientific literature. Please replace this probably wrong name with correctly one.

Line 219. We also noticed. In scientific article is not advisable this grammatical form.

Lines 224-225. Verbenone is camphor, celery and menthol tasting compounds. What does it mean?

Line 232. They have grown. Who?

Line 253. For the PCA and HCA the EOs components with percentage less than 3% were excluded.   Why?

There are not explanation or demonstration or experiments that show the no-importance of these variables. It is not possible a-priori to know what is better to include and what is better to exclude. Please to justify in a rationale manner the exclusion of these compounds.

Line 281. The analysis explained about 99.21%. 99.21% is the cumulative explained variance by the first two components, and not by the analysis.

Line 283. The first principal component…is positively correlated with…In the PCA technique the components explain the features but are not correlate with them. The features explained by the same component are correlate among them, and in the same time the components are orthogonal them to guarantee the independence among the variable explained by them. In figure 5 was reported the loading plot and on the axes there are the variables weight on the analysed component, not the correlation coefficient.

Lines 284. They were predominant in L1/L2 samples (Groupe B, Figure 5). In figure 5 there is the loading plot but to declare that a group of compounds characterizes a list of samples is needed the PCA score plot, here absent. Is not satisfactory to add on the loading plot the hypothetical samples organization (hypothetical because it is not the PCA results)

Line 288. How does the second component explain about 59% of the data and the first one 40.21%?

In PCA technique the first component explains a maximum quantity of total data and is impossible that the second one explains a major percentage than the first one.

Figure 5. Please to review the axis name.

Where is the Figure 7??

Lines 373-381. It is very difficult to understand this project step. What is the aim of this PCA application?

In the figure 9 there are ‘PCA1’ and ‘PCA2’ why? Are these two different PCA? In this case is not correct to analyse together the results.

If not, but them are plots from the same PCA a most exhaustiveness explanation of the results is needed to clarify the aim and the evidences.

To review 3.2 and 3.3 sections, the English and the experiment descriptions are not appropriate.

Table 4. Please to review the table structure. What are the number in each cell?

Please to modify the columns name to render simpler the lecture of the data in the table.

“All Values differs at p<0.05” doesn’t need of the asterisk because is true for all values.

Please to add the references compound.

Table 5. To give specific the concentrations used is better here to use the apexes, not asterisks.

C1a, C2b and C3c.

“All Values differs at p<0.05” doesn’t need of the apexes because is true for all values.

Please to change antibiotics with different expression, for example Control compounds or reference compounds.

Please to modify the size table.

Table 6. Please to add the control compound.

Please to insert the MICs and the MBCs in two different columns, splitting the three columns in six columns.

Tested microorganism (strains)

S.aureus

E.coli

P.aeruginosa

MIC

MBC

MIC

MBC

MIC

MBC

The abbreviation MIC is MIC, not CIM, and MBC is MBC, not CMB!!

Please to add the reference compound.

Reviewer 3 Report

The manuscript entitled "Chemical composition and antibacterial activity of Lippia multiflora Moldenke essential oil from 3 different regions of Angola" was reviewed. This is an important study. The Authors suggest that Angolan Lippian multiflora Moldenke has antibacterial properties and could be a potential source of antimicrobial agents for the pharmaceutical and food industry.

The appropriate methods were used. The Authors described results in details. The paper is very interesting. I recommended this article for publication without revision.

Author Response

Thank you for your comment.

Round 2

Reviewer 2 Report

Point and response 20. In this way, deleting a variables from the PCA technique you are altering the description of the samples, and it is not possible to understand what essential oils component is truly determinant for a specific sample.

So, to better interpreter PCA results and relative plots the variables elimination is not a right way. To clarify the results and plots is to useful to establish a loading value cut-off to exclude from the plots the feature with a low importance on specific Principal Component.

Point and response 22. I’m so sorry, but I think that some concepts are not clear.

What the values in table 3 (Component matrix generated for PCA) are? Loading values? If yes these values represent each essential oil chemical component weight in each principal component elaboration and not a correlation coefficient with these.

It is not clear why the table of Pearson’s correlation coefficients was showed to explain the PCA results.

Please, to review the PCA concepts.

Point and response 23. In the revised manuscript was not showed the score plot, but only the loading plot in which were modified the axes name (Component2 à F2 and Component1 à F1), this is not satisfactory to analyse and to interpret PCA results.

What do F1 and F2 mean?

Again, please to insert the score plot and modify the PCA interpretation.

Point and response 27. I’m sorry, but the PCA application is not clear. The two plots derived from the same PCA application? The plot where there are the bacteria is a loading plot? The didascalies PCA(1) and PCA(2) are confuse and don’t help the readers to understand the concepts.

Author Response

Response to Reviewer 2 Comments (Round 2)

Point 1: Point and response 20. In this way, deleting a variable from the PCA technique you are altering the description of the samples, and it is not possible to understand what essential oils component is truly determinant for a specific sample.

So, to better interpreter PCA results and relative plots the variables elimination is not a right way. To clarify the results and plots is to useful to establish a loading value cut-off to exclude from the plots the feature with a low importance on specific Principal Component.

Point and response 22. I’m so sorry, but I think that some concepts are not clear.

What the values in table 3 (Component matrix generated for PCA) are? Loading values? If yes these values represent each essential oil chemical component weight in each principal component elaboration and not a correlation coefficient with these.

It is not clear why the table of Pearson’s correlation coefficients was showed to explain the PCA results.

Please, to review the PCA concepts.

Response 1: All PCA and HCA parameters have been changed. All chemical compounds of the oils were used in the analysis. We have set the PCA to Eigenvalue cut-off equal to 1.00. Three PCs have been selected out of the eight. The loading and score plots were thus determined. We carried out a cluster analysis (CD) using the Euclidean distance and the Unweighted Pair Group Method with Arithmetic Mean cluster algorithm. Heat-map (Correlation matrix) were then generated which assigned diversity of the 11 Lippia samples based on their essential oil components.

(please see now l.288 to l.371 of revised manuscript, and the supplementary table S1).

Point 2: Point and response 23. In the revised manuscript was not showed the score plot, but only the loading plot in which were modified the axes name (Component2 à F2 and Component1 à F1), this is not satisfactory to analyse and to interpret PCA results.

What do F1 and F2 mean?

Again, please to insert the score plot and modify the PCA interpretation.

Point and response 27. I’m sorry, but the PCA application is not clear. The two plots derived from the same PCA application? The plot where there are the bacteria is a loading plot? The didascalies PCA(1) and PCA(2) are confuse and don’t help the readers to understand the concepts.

Response 2: All PCA and HCA parameters have been changed. All chemical compounds of the oils were used in the analysis. We have set the PCA to Eigenvalue cut-off equal to 1.00. Three PCs have been generated. The loading and score plots were thus determined. We carried out a cluster analysis (CD) using the Euclidean distance and the Unweighted Pair Group Method with Arithmetic Mean cluster algorithm. Heat-map (Correlation matrix) was then established, which made it possible to differentiate the oils according to their antibacterial effectiveness.

(please see now l.461 to l.642 of revised manuscript, and the supplementary Table S2).